# A Matter of Degrees: A Systematic Review of the Ergogenic Effect of Pre-Cooling in Highly Trained Athletes

**DOI:** 10.3390/ijerph17082952

**Published:** 2020-04-24

**Authors:** Miguel Ángel Rodríguez, José Víctor Piedra, Mario Sánchez-Fernández, Miguel del Valle, Irene Crespo, Hugo Olmedillas

**Affiliations:** 1Department of Functional Biology, Universidad de Oviedo, 33006 Oviedo, Spain; miguerguez95@gmail.com (M.Á.R.); jvpiedra@hotmail.com (J.V.P.); info@fisiomariosanchez.com (M.S.-F.); icreg@unileon.es (I.C.); 2Department of Cellular Morphology and Biology, Universidad de Oviedo, 33006 Oviedo, Spain; miva@uniovi.es; 3Institute of Biomedicine, Universidad de León, 24071 León, Spain; 4Health Research Institute of the Principality of Asturias (ISPA), 33011 Oviedo, Spain

**Keywords:** cooling, hyperthermia, thermoregulation, athletic performance

## Abstract

The current systematic review evaluated the effects of different pre-cooling techniques on sports performance in highly-trained athletes under high temperature conditions. PubMed/MEDLINE, EMBASE, Web of Science, CENTRAL, Scopus, and SPORTDiscus databases were searched from inception to December 2019. Studies performing pre-cooling interventions in non-acclimatized highly-trained athletes (>55 mL/kg/min of maximal oxygen consumption) under heat conditions (≥30 °C) were included. The searched reported 26 articles. Pre-cooling techniques can be external (exposure to ice water, cold packs, or cooling clothes), internal (intake of cold water or ice), or mixed. Cooling prior to exercise concluded increases in distance covered (1.5–13.1%), mean power output (0.9–6.9%), time to exhaustion (19–31.9%), work (0.1–8.5%), and mean peak torque (10.4–22.6%), as well as reductions in completion time (0.6–6.5%). Mixed strategies followed by cold water immersion seem to be the most effective techniques, being directly related with the duration of cooling and showing the major effects in prolonged exercise protocols. The present review showed that pre-cooling methods are an effective strategy to increase sports performance in hot environments. This improvement is associated with the body surface exposed and its sensibility, as well as the time of application, obtaining the best results in prolonged physical exercise protocols.

## 1. Introduction

Nowadays, sport competitions take place in a great diversity of geographical areas characterized by hot environments, such as the 2019 International Association of Athletics Federations (IAAF) World Championships in Doha, the 2020 Olympic Games in Tokyo, and the 2022 Federation International Football Association (FIFA) World Championship in Qatar. This situation constitutes a major challenge for athletic trainers and medical staff due to the impact that heat causes in athletes, mainly in endurance, racket, and team-sport disciplines. The practice of moderate/high-intensity exercise produces large amount of energy, which is eliminated as heat with an associated increase of central temperature [1]. Skin blood flow and sweat rate increments are crucial thermoregulatory mechanisms that favor heat loss [2], although when environmental conditions are extreme, these adjustments are disturbed and cannot avoid the elevation of core body temperature [3]. In this regard, hyperthermia is known to reduce physical and athletic performance [4,5], altering cardiovascular function and leading to both peripheral and central fatigue [6]. Furthermore, exertional heat stroke may occur when core temperature reaches 40 °C and the subject begins to suffer changes in mental status [7]. Nonetheless, highly trained endurance athletes respond physiologically as if they were already heat acclimatized [8], and present less adaptive potential in comparison with moderately trained athletes or untrained subjects [9]. In fact, a study performed in elite cyclists competing in heat showed that they were able to reach core temperatures of 40 °C and above without heat illness [10].

Bearing in mind the particular circumstances in which highly trained athletes compete and in looking for a major performance, sport scientists have put all their effort to find cooling techniques to reduce central temperature and delay the onset of fatigue [11]. Thus, athletes can implement cooling before or during competition to facilitate heat dissipation and increase heat storage capacity, prolonging the time in which exercise intensity can be maintained before reaching a critical top limit [12].

To date, none of the reviews on this topic have focused on highly-trained athletes (>55 mL/kg/min of maximal oxygen consumption (VO_2_max)) under heat stress conditions (>30 °C) [11,13,14]. This aspect is extremely necessary in preparation for attending to the sports events that will take place in the near future. Therefore, the purpose of this systematic review was to summarize the current scientific evidence in relation to the effectiveness of pre-cooling strategies in highly-trained athletes exercising/competing in high temperature environments.

## 2. Materials and Methods

This systematic review followed the Preferred Reporting Items for Systematic Reviews and Meta-Analyses (PRISMA) statement [15].

### 2.1. Search Strategy

All studies were identified through a search on electronic databases, including PubMed/MEDLINE, EMBASE, Web of Science, CENTRAL, Scopus, and SPORTDiscus, from inception until December 2019. Reference lists of the included articles were also searched for additional references. Searches were restricted to the English language, without limitations on dates of publication. Details of the search strategy for PubMed/MEDLINE are shown in Appendix A.

### 2.2. Eligibility Criteria

Studies were included if they fulfilled the following inclusion criteria: (1) cooling intervention applied before exercise; (2) existence of a control condition (without cooling intervention) through a randomized crossover design; (3) ambient temperature ≥30 °C; (4) highly-trained athletes (>55 mL/kg/min of VO_2_max or, in case this value was not mentioned, the clear specification by the authors that the athletes were trained); (5) athletes not acclimatized to heat; and (6) measurement of sports performance. Studies were excluded if they were incomplete (e.g., abstracts), if outcome measures were based on non-performance parameters (e.g., physiological markers), and/or if their design was different from randomized crossover trial. 

### 2.3. Study Selection

Articles identified by the search strategy were screened independently by two authors for the inclusion criteria using the title and abstract and then the full-text copies. Discrepancies over article inclusion were settled through discussion with a third reviewer until consensus was reached. Data were extracted independently by two investigators, and involved the following items: data on study source, study design, sample size, characteristics of the participants, ambient conditions (temperature and relative humidity), exercise protocol, technique and protocol of pre-cooling, variation in core temperature (Tc) between periods at the end of exercise, and final outcomes of the interventions. The main outcomes were total distance covered, power output, completion time, time to exhaustion, and work.

### 2.4. Quality Assessment

The quality of the studies was assessed by two researchers using the PEDro scale for this purpose [16]. This scale is based on 11 items, the first of which refers to external validity, and the remaining 10 to internal validity and the presentation of the statistical analysis. The assessment was nevertheless scored out of 10, as the first item was not taken into account for this purpose. For each item whose criterion was met, one point was awarded, whereas no points were given if the item was not fulfilled [17]. The relationship between the score and the quality of the study was on the following basis: excellent quality (9 or 10 points), good quality (6 to 8 points), fair quality (4 or 5 points), and poor quality (fewer than 4 points). 

## 3. Results

### 3.1. Studies Included

The search strategy retrieved 2845 records. After duplicates were removed, 1253 studies were excluded from the review process and 1525 were excluded after title and/or abstract analysis; 67 full-text copies of the remaining studies were obtained and subjected to further evaluation. After reading full-text copies, 41 studies were excluded from this review and 26 articles meeting the eligibility criteria were included for qualitative analysis [18,19,20,21,22,23,24,25,26,27,28,29,30,31,32,33,34,35,36,37,38,39,40,41,42,43] (Figure 1).

### 3.2. Participant Characteristics 

Overall, 240 subjects were included in the qualitative analysis. The number of participants ranged from 6 [41] to 20 [25]. All the studies assessed only men, except three which used a mix-sex sample [20,21,26] and two studies that included only women [44,45]. In total, 227 males and 13 females took part in the studies. The average age of participants was 24.6 years (ranging between 19.9 [32] and 34.8 [22]), and the VO_2_max varied from 55.7 [33,38] to 71.6 mL/kg/min [40]. Regarding sport modality, studies included a wide variety of athletes, both individual (cyclists [18,19,24,30,37,38,39,40,41,42], runners [20,21,22,43], triathletes [23,24,25,42], and tennis players [26]), and team-sport players (covering cricket [27,28,29], soccer [31], lacrosse [32], and other non-specified team sports [33]). Furthermore, three studies included volunteers who recreationally practice various sport activities [34,35,36]. Table 1 summarizes the main results of the selected studies.

### 3.3. Intervention Characteristics

All the studies analyzed the acute effects of pre-cooling on sports performance and compared it with non-pre-cooling controls. A total of 20 studies evaluated external techniques through 28 interventions [18,19,20,21,22,24,25,26,27,28,29,31,32,33,37,38,39,40,41,43], 8 assessed internal techniques (intake of ice or cold water) [23,30,31,34,35,36,42,43], and 3 investigated mixed strategies (a combination of external and internal methods) [31,40,43].

Two studies measured time to exhaustion (TTE) [34,36], three analyzed work [18,27,39], one evaluated mean peak torque [28], and one assessed cricket parameters (ball speed, accuracy, and total run-up speed) [29]. Pre-cooling strategies studied were:External: cold water immersion (CWI) [21,33,37,38], ice packs [22,31], iced towels [27], cooling gloves [39], ice vest/jacket [18,20,39,41], cooling garment [19], cold water over the head [43], and diverse combinations of the above techniques [24,26,27,28,29,32,40,41].Internal: crushed ice ingestion [23,30,31,34,35,36,42] and oral rehydration [43].Mixed: a combination of external and internal techniques [31,40,43].

### 3.4. Outcome Measures

#### 3.4.1. External Method Cooling vs. Non-Cooling Strategies

The use of external cooling devices showed increases in distance covered (3.6–13.1%), mean power output (0.9–4.5%), work (0.1–8.5%), and mean peak torque (10.4–22.6%), in addition to reductions in completion time (0.5–5.8%) when compared with non-cooling strategies.

#### 3.4.2. Internal Method Cooling vs. Non-Cooling Strategies

Internal cooling elicited increases in distance (1.5%), mean power output (6.2%), and TTE (19–31.9%), as well as reductions in completion time (0.6–6.5%) in comparison with non-cooling control groups.

#### 3.4.3. Mixed-Method Cooling vs. Non-Cooling Strategies

The combination of internal and external strategies was effective in increasing mean power output (3%) and reducing completion time (1.3–5.1%).

#### 3.4.4. Inter-Group Comparisons

Those studies comparing mixed-method cooling vs. internal and/or external strategies did not conclude any significant between-group effects, although there was a trend in favor of mixed methods [31,40,43].

### 3.5. Quality Assessment 

Regarding methodological quality, the risk of bias of the included studies is shown in Table 2. Due to their crossover design, none of the studies could conceal allocations and blind participants from the interventions. Moreover, none of them blinded therapists or assessors. Overall, the quality of the studies attained an average of 4.9 in the PEDro scale, which corresponded to a fair quality.

## 4. Discussion

Pre-cooling techniques seem to be an effective strategy to enhance sports performance in hot environments. Thus, improvements in terms of time, distance covered, work and power output, and increasing TTE have been shown after applying cooling strategies prior to acute exercise trials. 

With regard to external pre-cooling, a combination of methods appears to yield better results than the use of an isolated technique. A reduced mean completion time was observed after a combination of CWI with an ice vest/jacket, compared with ice vest or jacket alone [41]. Minett et al. [27] found that 20 min of whole-body pre-cooling was more effective than head and hands cooling in intermittent sprint bouts, and even larger differences were observed compared to head refreshing alone. However, Maroni et al. [39] did not show any differences in performance when comparing a hand-cooling glove technique with a cooling jacket or a combination of both during high intensity prolonged repeated-sprint efforts in cyclists. Despite this circumstance, all techniques improved thermal sensation when they were applied in isolation. Because none of the cooling techniques assessed were able to reduce Tc values between all trials [39], this factor might play an important role in athletic performance. The key to reducing the Tc could reside in the parts of the body exposed to cooling devices, suggesting that head and neck are more decisive than the distal parts. Neck is located near the thermoregulatory center, which receives and integrates afferent thermal inputs and coordinates the efferent response to the periphery [46]. Both neck and head present 2-5 times higher alliesthesial thermosensitivity than any other segment during moderate cooling [47] and dominate whole-body temperature perception despite constituting only a small portion of the total skin surface area [48]. In this regard, cooling the neck was observed to significantly extend TTE in untrained individuals performing a treadmill time-trial [49]. Furthermore, the distance covered increased when endurance-trained subjects were cooled prior and during a similar exercise protocol [50]. In addition to this, the specificity in the physiological assessment of performance may have a key role in these findings [51], as cricket players are more familiar with sprints [27] than cyclists, who are not used to performing repeated short-duration sprints (15 s) on a cycle-ergometer [39]. However, the hypothesis that cycling may not benefit from pre-cooling was discarded, as there are a considerable number of studies that have reported positive results using cooling devices prior to simulated cycling tests [18,19,24,25,30,37,38,40,41].

Concerning individual techniques, CWI showed the best results among the external strategies in terms of sports performance, offering significant increases in total distance covered [21,33,37], as well as reductions in completion time [38]. Although these previous results support the fact that cooling a larger body surface area improves performance to a greater extent, the application of ice packs alone on quadriceps and hamstrings also yielded significant positive effects in completion time after performing a 5000m treadmill time-trial [22]. Moreover, cooling hamstrings and quadriceps improved power and work following a sprint-based protocol on a cycle-ergometer [52], even though local cooling slows enzyme activity and nerve conduction, as well as reducing the rate of force development [53]. This reduction in contractile speed shown by the cooled muscles [54] especially affects short and intense efforts, as a consequence of its negative impact on the fast twitch fiber recruitment, essential for an optimal sprint performance [55]. Therefore, although further research is needed, it may be counterproductive to cool lower-limb muscles prior to sprint protocols, as the high pre-activation required by the hamstrings just before ground contact to produce maximum acceleration [56] could be harmed. 

Furthermore, and inversely related with intense efforts, prolonged exercise alters thermal homeostasis to a greater extent than shorter exercise protocols, responding better to pre-cooling and highlighting the importance of exercise duration on its effectiveness [57]. Nonetheless, it is worth mentioning that cooling effect may be limited and tends to decrease when exercise exceeds 60 min [11], and thus including the intervention process during exercise breaks (if exist) can help to maintain its effectiveness.

Regarding internal strategies, the ingestion of ice crushed before exercising managed to reduce completion time in both a 40km time-trial [30] and in the running phase of an Olympic-distance triathlon [23], as well as increases in distance covered on a treadmill [35]. Nevertheless, the timing of ice ingestions may be important for enhancing performance. In this regard, a significantly higher TTE was observed when the rest interval between the end of the ice intake and the beginning of the exercise protocol was more prolonged (20 min vs. 5 min) [58]. These results encourage research on the most appropriate time of ingestion in relation with the beginning of the competition to enhance the ergogenic effects of this technique.

Muñoz et al. [43] noted that oral rehydration (7 °C) each 10 min before completing a 5000 m treadmill could reduce a 4.7% time in the race test, although in a non-significant manner. Despite the short duration of the test, this fact might highlight the importance of an adequate pre-hydration level on sports performance, as has been previously described in long-distance runners [59] and cyclists [60,61]. Nevertheless, the deleterious effect of hypohydration is still in doubt, as well-trained cyclists subjected to a dehydration up to −3% did not impair their performance on a 25 km time-trial in hot conditions (33 °C, 40% relative humidity (rh)) [62]. These results may be taken into account, as they have been obtained under real conditions (blinding the cyclists to their hydration status and with a facing air speed at a rate near that of field-based forward cycling speed).

As expected, when internal and external strategies are combined, the effectiveness is enhanced in comparison with the application of both separately. The intake of crushed ice ingestion was significantly more effective than CWI in terms of power output when both were combined separately with a cooling jacket [40], and adding ice slushy to a cooling jacket improved power and work values compared to the external technique alone in team sport athletes [63]. Therefore, it seems that mixed strategies are the best alternative to boost athletic performance under heat stress conditions, although further research studying practical applicability is needed to establish the most feasible techniques. 

Despite all the above, this review focused on athletes non-acclimatized to heat, and thus those who perform an adaptation period before competing might not obtain further positive results after pre-cooling aid, as has already been suggested. In this regard, no beneficial effects of pre-cooling were observed in terms of performance in acclimatized amateur trained runners [64]. However, 2 weeks of acclimatization offered similar results compared with cool conditions in terms of time when trained cyclists were subjected to a prolonged time trial, whereas they showed a decrease in power output in the non-acclimatized period [65]. Accordingly, it has been concluded that crushed ice ingestion prior to exercise offered identical improvements in endurance cycling performance compared with a 12 day acclimatization training program [66], being a cost-effective, time-efficient alternative if the acclimatization is not possible. Apart from this, the individual characteristics of each athlete constitute a differential factor concerning their response to heat. The skin temperature of athletes exposed to heat raises at a slower rate, as they sweat earlier than less-trained individuals [67]. This fact has led to suggestions that the application of cooling devices before exercise does not provide any beneficial effects [12]. However, this hypothesis may depend on athletes’ morphology. Larger individuals have a lower increase of body temperature, whereas subjects with higher aerobic fitness tended to accumulate more heat for the same relative exercise intensity during cycling in hot conditions, as heat production exceeds its elimination [68].

### Future Directions on Heat Management in Athletes

Sport competitions are currently taking place in non-traditional hot countries, and thus the implementation of cooling techniques before, during, and after competitions will become even more important for athletes to cope with the heat. We have concluded that pre-cooling is effective in improving performance parameters in highly trained athletes. However, no solid evidence exists about the implication of cooling strategies on short and intense efforts, as we have mentioned above. Furthermore, it is also important to find the most suitable pre-cooling technique for each sports practice or specific context, as it may occur that the most effective technique in absolute terms is not the most adequate in certain circumstances. In addition, and parallel to these techniques, the development of technology as wearable biosensors capable of monitoring the thermal state and the response to physical exertion in real time will provide additional advantages to medical staff to protect the health of athletes [69,70].

## 5. Conclusions

Pre-cooling methods are an effective strategy to reduce core body temperature prior exercise sessions in heat environments. These methods boost athletic performance, generally increasing power, work, covered distance, and time to exhaustion, as well as reducing completion time. Mixed techniques (a combination of internal and external strategies) seem to be the most adequate option to enhance the benefits of this practice, closely followed by cold water immersion, highlighting the influence of the body surface exposed. Additionally, prolonged physical exercise protocols have evidenced better results from the application of pre-cooling than short bouts. 

## Figures and Tables

**Figure 1 ijerph-17-02952-f001:**
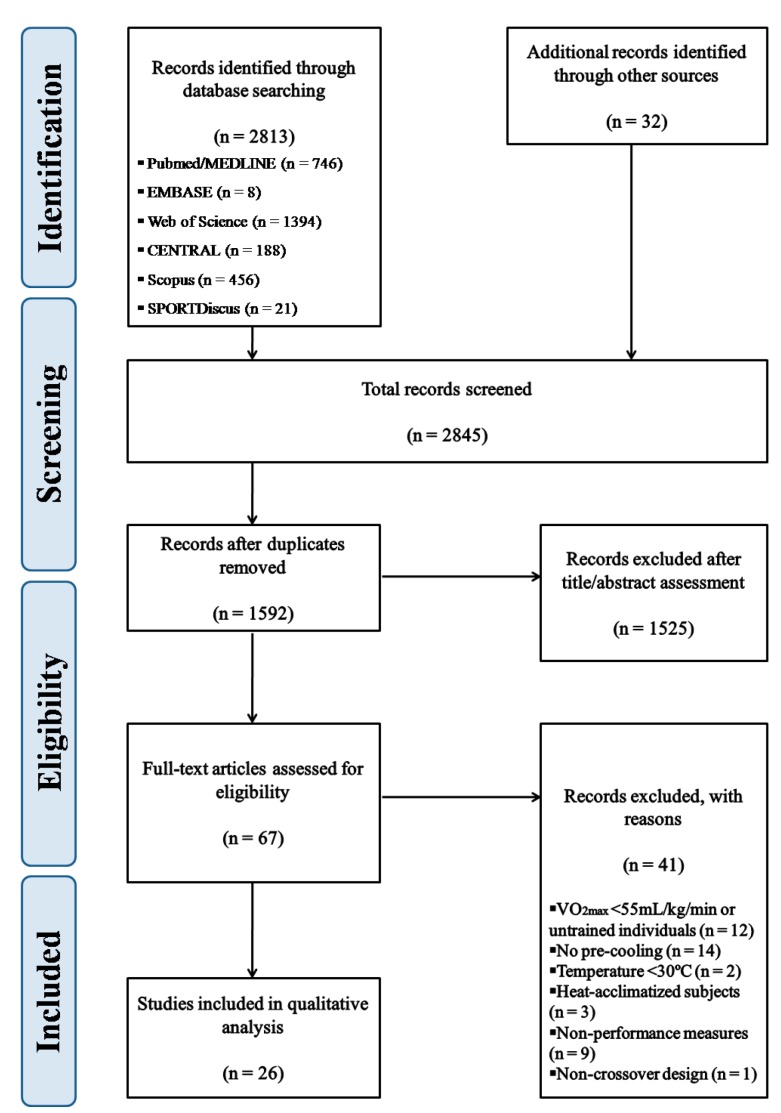
Flow diagram of study selection process.

**Table 1 ijerph-17-02952-t001:** Summary of the included studies.

Study	N ▪Sex ▪Age ▪VO_2_max ▪Sport	EC ▪T ▪RH	Exercise Protocol (Time/Distance, Timing and Type)	Pre-Cooling Strategy		Main Results (vs. CON)		Between-Group Effects (Inter-COOL)
				App	Protocol	ΔTc Max #	(1) **Distance** (2) **Mean power output** (3) **Completion time** (4) **Time to exhaustion** (5) **Work** (6) **Mean peak torque** (7) **Cricket-specific measures**	(1) **Distance** (2) **Mean power output** (3) **Completion time** (4) **Time to exhaustion** (5) **Work** (6) **Cricket-specific measures**
Aldous et al. [31]	8♂ 22 ± 3 56 ± 9 Soccer	30.7 51	90′ 45′/15′rec/45′ TM	INT	Ice slurry ingestion (7.5g/kg pre @ −0.1 °C and 3.75g/kg rec)	−0.1	(1) 0%	No significance
				EXT	Ice packs on quadriceps and hamstrings @ −14 °C	0	(1) 0%	
				MIX	Ice slurry ingestion + ice packs	−0.2	(1) 0%	
				CON	Water ingestion (21 °C)			
Arngrïmsson et al. [20]	9♂, 8♀ 22.7 ± 3.3 62.3 ± 4.4 Running	32 50	5000 m TT TM	EXT	38′ ice vest	−0.2	(3) ↓1.1% (*p* < 0.05)	
				CON	No COOL			
Booth et al. [21]	5♂, 3♀ 26.7 ± 1.7 63.1 ± 0.1 Running	32 60	30′ TT TM	EXT	CWI (10′ @ 28–29 °C + 50′ @ 23–24 °C)	−0.8	(1) ↑4.2% (*p* < 0.05)	
				CON	No COOL			
Duffield et al. [18]	7♂ 20.2 ± 2.2 ND Cycling	30 60	80′ sprints 3 × 15′/5′-10′ rec CE	EXT	Ice jacket (pre and rec times)	−0.1	(2) ↑2.4% (5) ↑4.2%	
				CON	No COOL			
Duffield et al. [26]	6♂, 2♀ 20.8 ± 1.5 ND Tennis	35 55	Running (on-court tennis movement drills) 5 × 5′/2′ rec	EXT	Ice vest + cold towels on head/neck/legs (20′ @ 5 °C) + cold compression garment (10′ @ 5 °C)	−0.2	(1) ↑4.6%	
				CON	No COOL (passive seating)			
Duffield et al. [32]	7♂ 19.9 ± 1.4 ND Lacrosse	32.4 44	30′ sprints 4 × 5′/2′ rec Lacrosse field	EXT	20′ COOL vest + cold towels in neck (3 °C) + ice packs on quadriceps	−0.5	(1) ↑7.7% (*p* = 0.05)	
				CON	No COOL			
Faulkner et al. [19]	10♂ 25.1 ± 6.1 61.3 ± 4.3 Cycling	35 51	60′ (75% Wmax) CE	EXT	40′ COOL garment cold water (14.2 °C)	0	(2) ↓2.1% (3) ↓3.6%	(2) No significance (3) ↑2.6% in favor of COOL frozen water (vs. COOL cold water) (*p* < 0.05)
				EXT	40′ COOL garment water frozen	−0.1	(2) ↑4.5% (*p* < 0.05) (3) ↓5.8% *(p* < 0.05)	
				CON	No COOL			
Gerrett et al. [35]	12♂ 30.4 ± 3.4 58.5 ± 8.1 Various sport activities	30.9 41.1	2 × 31′ self-pace intermittent protocols TM	INT	30′ ice slurry ingestion (7.5 g/kg @ 0.1 °C)	−0.2	(1) ↑1.5%	
				CON	Water ingestion (23.4 °C)			
Ihsan et al. [30]	7♂ 27.7 ± 3.1 ND Cycling	30 75	40 km TT CE	INT	30′ crushed ice ingestion (6.8 g/kg)	−0.4	(2) ↑6.9% *(p* = 0.06) (3) ↓6.5% *(p* = 0.049)	
				CON	No COOL			
Katica et al. [24]	8♂ 25 ± 3 57.8 ± 5.0 Cycling/Triathlon	35 43.8	16.1 km TT CE	EXT	20′ head and neck ice wraps + ice vest	0.1	(2) ↑1.8% (3) ↓3.4% (*p* = 0.04)	
				CON	No COOL			
Kay et al. [37]	7♂ 23.7 ± 2.1 64.5 Cycling	31 60	30′ TT CE	EXT	CWI (10′ @ 29.7 °C + 50′ @ 8–11 °C)	−0.3	(1) ↑6.0% (*p* < 0.05)	
				CON	No COOL			
Lee et al. [34]	8♂ 22 ± 4 57.8 ± 5.6 Various sport activities	35 60	Cycle to exhaustion	INT	Cold liquid ingestions (3 × 300 mL @ 4 °C)	−0.5 (*p* < 0.001)	(4) ↑31.9% (*p* < 0.001)	
				CON	Warm liquid ingestion (3 × 300 mL @ 37 °C)			
Maia-Lima et al. [38]	8♂ 20 ± 1 55.7 ± 7.88 Cycling	35 68	30 km TT CE	EXT	CWI (10 × (3′@ 24 °C) and 3′ out of the bath)	−0.9 (*p* < 0.05)	(2) 0% (3) ↓3.4% (*p* < 0.05)	
				CON	No COOL			
Maroni et al. [39]	10♂ 21.1 ± 3.3 65.7 ± 10.7 Cycling	35 68	43′ sprints (cycling race simulation protocol) CE	EXT	30′ hand-COOL glove	−0.3	(2) ↑1.0% (5) ↑0.1%	No significance
				EXT	30′ COOL jacket	−0.3	(2) ↑3.1% (5) ↑3.0%	
				EXT	30’ hand-COOL gloves + COOL jacket	−0.5	(2) ↑0.9% (5) ↑0.8%	
				CON	No COOL			
Minett et al. [27]	10♂ 20.9 ± 2.6 ND Cricket	33 33	70′ sprints 2 × 35′/15′ rec Running track (cricket simulation)	EXT	Head COOL (iced towel soaked in water at 5 °C)	0	(1) ↑5.8% (5) ↑0.3%	(1) ↑6% and 7% in favor of whole-body COOL (vs. head + hand-COOL and hand-COOL, respectively)
				EXT	Head + hand-COOL (iced towel soaked in water at 5 °C in head and cold water immersion @ 9 °C in hands)	−0.1	(1) ↑6.9% (*p* < 0.05) (5) ↑8.2%	
				EXT	Whole-body COOL (iced towel on head and neck @ 5 °C, cold water on hands @ 9 °C, ice vest on torso and ice packs on quadriceps @ −20 °C −20′ + 5′ rec)	−0.4	(1) ↑13.1% (*p* < 0.05) (5) ↑8.5%	
				CON	No COOL			
Minett et al. [28]	8♂ 21.5 ± 2.7 ND Cricket	33 34	70′ sprints 2 × 35′/15′ rec Running track (cricket simulation)	EXT	10′ + 5′ rec of whole-body COOL (iced towel on head and neck @ 5 °C, cold water on hands @ 9 °C, ice vest on torso and ice packs on quadriceps @ −20 °C)	−0.1	(1) 0% (6) ↑10.4%	(1) ↑4.5% in favor of 20′ COOL (vs. 10′ COOL) (*p* = 0.03) (6) ↑11.1% in favor of 20′ COOL (vs. 10 ′COOL) (*p* = 0.03)
				EXT	20′ + 5′ rec of whole-body COOL (iced towel on head and neck @ 5 °C, cold water on hands @ 9 °C, ice vest on torso and ice packs on quadriceps @ −20 °C)	−0.4 (*p* < 0.05)	(1) ↑4.7% (*p* = 0.01) (6) ↑22.6% (*p* = 0.05)	
				CON	No cooling			
Minett et al. [29]	10♂ 23 ± 8 ND Cricket	31.9 63.5	6-over bowling spell 10 m walking/20 m sprinting	EXT	20′ towel on head, neck and shoulders (5 °C) + ice vest (−20 °C), cold water on non-bowling hand (9 °C) and ice-packs on quadriceps (−20 °C)	−0.3	(7) ↑0.3% ball speed (*p* = 0.63) ↓2.5% accuracy (*p* = 0.76) ↓1.0% total run-up speed (*p* = 0.66)	
				CON	No COOL			
Muñoz et al. [43]	10♂ 25 ± 4 60.2 ± 5.4 Running	33 30	5000 m TT after 90′/30%VO_2_max TM	INT	Oral rehydration (7 °C) each 10′	−0.2	(3) ↓4.7%	No significance
				EXT	Cold water over the head (7 °C) each 10′	−0.1	(3) ↓3.8%	
				MIX	Oral rehydration (7 °C) + cold water over the head (7 °C) each 10′	−0.2	(3) ↓5.1%	
				CON	No COOL			
Quod et al. [41]	6♂ 28 ± 4 71.4 ± 3.2 Cycling	34 41	40′ @ 75% WmaxCE	EXT	CWI (5′ @ 29 °C + 25′ @ 24 °C) + ice jacket (40′)	−0.2 (*p* = 0.004)	(2) ↑3.8% (3) ↓3.8% (*p* = 0.009)	(3) ↓2.4% in favor of the combined treatment (vs. COOL jacket) (*p* = 0.06)
				EXT	40′ COOL jacket	0	(2) ↑1.6% (3) ↓1.5% (*p* = 0.35)	
				CON	No COOL			
Randall et al. [22]	8♂ 34.8 ± 4.4 65.5 ± 3.9 Running	32 49	5000 m TT TM	EXT	30′ ice packs on quadriceps and hamstrings	0	(3) ↓6.0% (*p* < 0.01)	No significance
				EXT	30′ ice vest	−0.2	(3) ↓3.2%	
				CON	No COOL			
Ross et al. [40]	11♂ 33 ± 5.1 71.6 ± 6.1 Cycling	32–35 50–60	46.4 km TT CE	MIX	30′ crushed ice ingestion (14g/kg) + iced towels on torso and legs	ND	(2) ↑3.0% (*p* = 0.04) (3) ↓1.3% *(p* = 0.08)	No significance
				EXT	CWI (10′ @ 10 °C) + 20′ ice jacket	ND	(2) ↑1.1% *(p* = 0.43) (3) ↓0.5% *(p* = 0.53)	
				CON	Cold water (4 °C) ingestion *ad libitum*			
Schmit et al. [25]	13♂ 31 ± 4 64.9 ± 6.9 Triathlon	35 50	20 km TT CE	EXT	20′ ice vest	ND	(2) ↑4.0% (3) ↓1.3%	
				CON	No COOL			
Siegel et al. [36]	10♂ 28 ± 6 56.4 ± 4.7 Various sport activities	34 54.9	Run to exhaustion TM	INT	30′ ice slurry ingestion (7.5 g/kg @ −1 °C)	0.3 (*p* = 0.001)	(4) ↑19% *(p* = 0.001)	
				CON	Cold water ingestion (4 °C)			
Skein et al. [33]	10♂ 28 ± 2.7 55.7 ± 7.9 Team-sports	31 33	50 sprints (1 sprint/’with 1′ rec each 10′) Running track	EXT	CWI (15′ @ 10 °C)	−0.2	(1) ↑3.6% (*p* < 0.05)	
				CON	No COOL			
Stanley et al. [42]	10♂ 30 ± 5.0 60.0 ± 7.7 Triathlon/Cycling	34 60	75′ cycling @ 60% PPO + 50′ seated rec + 30′ performance trial) CE	INT	Ice slushy ingestion (−0.8 °C prior to performance trial)	−0.4 (*p* = 0.001)	(3) ↓0.6% *(p* = 0.263)	
				CON	Liquid ingestion (18.4 °C prior to performance trial)			
Stevens et al. [23]	9♂ 29.1 ± 3.6 61.7 ± 4.7 Triathlon	32–34 20–30	Triathlon (Olympic distance)	INT	Crushed ice ingestion (10g/kg) during 17′–45′ of cycling phase	−0.8	(3) ↓2.5% (running phase) *(p* = 0.03)	
				CON	Fluid ingestion (32–34 °C)			
				CON	No collar			

App: approach; CE: cycle ergometer; CON: control; COOL: cooling; CWI: cold water immersion; EC: environmental conditions; EXT: external strategy; HRmax: maximum heart rate; INT: internal strategy; km: kilometre; m: meter; Max: maximum (at the end of the exercise protocol); MIX: mixed strategy; mL: millilitre; N: sample; ND: no data; PPO: peak power output; rec: recovery; RH: relative humidity (%); RPE: rating of perceived exertion; T: temperature (°C); Tc: core temperature (°C); TM: treadmill; TT: time trial; VO_2_max: maximal oxygen uptake (mL/kg/min); VO_2_peak: oxygen uptake during peak exercise (mL/kg/min); vs.: versus; Wmax: maximal power; @: at; **#**: negative value favors COOL condition; ‘: minutes; ‘’: seconds; *: *p*: significance.

**Table 2 ijerph-17-02952-t002:** Quality assessment of the included studies.

Study	PEDro Score Distribution											
	1	2	3	4	5	6	7	8	9	10	11	Total PEDro Score
**Aldous et al.** [31]	─	●	─	●	─	─	─	●	─	─	●	4
**Arngrimsson et al.** [20]	─	●	─	●	─	─	─	●	─	●	●	5
**Booth et al.** [21]	─	●	─	●	─	─	─	●	─	●	●	5
**Duffield et al.** [18]	─	●	─	●	─	─	─	●	─	●	●	5
**Duffield et al.** [26]	─	●	─	●	─	─	─	●	─	●	●	5
**Duffield et al.** [32]	─	●	─	●	─	─	─	●	─	●	●	5
**Faulkner et al.** [19]	─	●	─	●	─	─	─	●	─	●	●	5
**Gerrett et al.** [35]	─	●	─	●	─	─	─	●	─	●	●	5
**Ihsan et al.** [30]	─	●	─	●	─	─	─	●	─	●	●	5
**Katica et al.** [24]	●	●	─	●	─	─	─	●	─	●	●	5
**Kay et al.** [37]	─	●	─	●	─	─	─	●	─	●	●	5
**Lee et al.** [34]	─	●	─	●	─	─	─	●	─	●	●	5
**Maia-Lima et al.** [38]	─	●	─	●	─	─	─	●	─	●	●	5
**Maroni et al.** [39]	─	●	─	●	─	─	─	●	─	●	●	5
**Minett et al.** [27]	─	●	─	●	─	─	─	●	─	●	●	5
**Minett et al.** [28]	─	●	─	●	─	─	─	●	─	●	●	5
**Minett et al.** [29]	─	●	─	●	─	─	─	●	─	●	●	5
**Muñoz et al.** [43]	●	●	─	●	─	─	─	●	─	●	●	5
**Quod et al.** [41]	─	●	─	●	─	─	─	●	─	●	●	5
**Randall et al.** [22]	─	●	─	●	─	─	─	●	─	●	●	5
**Ross et al.** [40]	─	●	─	●	─	─	─	●	─	●	●	5
**Schmit et al.** [25]	─	●	─	●	─	─	─	●	─	─	●	4
**Siegel et al.** [36]	─	●	─	●	─	─	─	●	─	●	●	5
**Skein et al.** [33]	─	●	─	●	─	─	─	●	─	●	●	5
**Stanley et al.** [42]	─	●	─	●	─	─	─	●	─	●	●	5
**Stevens et al.** [23]	─	●	─	●	─	─	─	●	─	●	●	5

1. Eligibility criteria; 2. Random allocation; 3. Concealed allocation; 4. Baseline comparability; 5. Blind subjects; 6. Blind therapists; 7. Blind assessors; 8. Adequate follow-up; 9. Intention-to-treat analysis; 10. Between-group comparisons; 11. Point estimates and variability. A “●” indicates a “yes” score, and a dash indicates a “no” score.

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
