# Peer review of "A Matter of Degrees: A Systematic Review of the Ergogenic Effect of Pre-Cooling in Highly Trained Athletes"

_ijerph, 2020, doi:10.3390/ijerph17082952_

Round 1

Reviewer 1 Report

  1. The opening sentence needs editing for clarity as suggested here: Nowadays sport competitions take place (and not part) in diverse geographical areas characterized by hot environments, such as the 2019 IAAF World Championships in Doha, the 2020 Olympic Games in 35
    Tokyo and the 2022 FIFA World Championship in Qatar etc
  2. Line 49 and 50 on page 2 require editing to make sense: "Bearing in mind the particular circumstances in which highly-trained athletes compete and looking for a major performance,
  3. Page 2, Line 53 ...it should be 'reaching' and not 'reach'
  4. Page 2, Line 57- add some words as shown: extremely necessary in preparation for attending to the sport
  5. Page 14, Line 147: a combination of methods appears to yield (delete- obtained) better result
  6. Page 14, Line  152-155- Edit this long sentence for clarity. I suggest you break it into two sentences rather than one: "However, Maroni et al [40] did not show any differences when compared (comparing) a hand-cooling glove technique with a cooling jacket or a combination of both during high intensity prolonged repeated sprint efforts in cyclists, despite all techniques improved thermal sensation when they were applied in isolation"
  7. Page 14, Lines 162-165 require editing for clarity: "In this regard, cooling the neck has been observed to extend significantly TTE in untrained individuals performed a treadmill time trial [50], as well as distance covered in endurance-trained subjects cooled prior and during a similar exercise protocol [51]".
  8. Page 14, Line 167- see change here   - "who are not used to performing repeated"
  9. Page 14, Line 169- add- "since there are considerable number"
  10. Page 14, Line 175- "has also yielded (in place of noted) significant"
  11. Page 15, Lines 241-243- edit for clarity. Better to break sentence into two shorter ones as suggested: "Pre-cooling methods are an effective strategy to reduce core body temperature prior to exercise sessions in heat environments. These methods boost athletic performance, generally increasing power, work, covered distance and time to exhaustion as well as reducing completion time"
  12. Good paper that adds value to literature on the area.

Author Response

Comments and Suggestions for Authors

The opening sentence needs editing for clarity as suggested here: Nowadays sport competitions take place (and not part) in diverse geographical areas characterized by hot environments, such as the 2019 IAAF World Championships in Doha, the 2020 Olympic Games in 35 Tokyo and the 2022 FIFA World Championship in Qatar etc

Done, we have changed it.

Line 49 and 50 on page 2 require editing to make sense: "Bearing in mind the particular circumstances in which highly-trained athletes compete and looking for a major performance,

We have rewritten it.

Page 2, Line 53 ...it should be 'reaching' and not 'reach'

Done.

Page 2, Line 57- add some words as shown: extremely necessary in preparation for attending to the sport

The words have been added.

Page 14, Line 147: a combination of methods appears to yield (delete- obtained) better result

This word has been deleted.

Page 14, Line 152-155- Edit this long sentence for clarity. I suggest you break it into two sentences rather than one: "However, Maroni et al [40] did not show any differences when compared (comparing) a hand-cooling glove technique with a cooling jacket or a combination of both during high intensity prolonged repeated sprint efforts in cyclists, despite all techniques improved thermal sensation when they were applied in isolation"

Done, we have modified it and we have broken the sentence into two sentences. “However, Maroni et al [40] did not show any differences in performance when comparing a hand-cooling glove technique with a cooling jacket or a combination of both during high intensity prolonged repeated-sprint efforts in cyclists. Despite this circumstance, all techniques improved thermal sensation when they were applied in isolation…” (lines 204-207).

Page 14, Lines 162-165 require editing for clarity: "In this regard, cooling the neck has been observed to extend significantly TTE in untrained individuals performed a treadmill time trial [50], as well as distance covered in endurance-trained subjects cooled prior and during a similar exercise protocol [51]".

Done, we have edited this sentence and we have divided it into two independent sentences. “In this regard, cooling the neck has been observed to extend significantly TTE in untrained individuals performed a treadmill time-trial [50]. Furthermore, the distance covered increased when endurance-trained subjects were cooled prior and during a similar exercise protocol…” (lines 215-217).

Page 14, Line 167- see change here   - "who are not used to performing repeated"

Thank you, the change has been made

Page 14, Line 169- add- "since there are considerable number"

Done.

Page 14, Line 175- "has also yielded (in place of noted) significant"

Done.

Page 15, Lines 241-243- edit for clarity. Better to break sentence into two shorter ones as suggested: "Pre-cooling methods are an effective strategy to reduce core body temperature prior to exercise sessions in heat environments. These methods boost athletic performance, generally increasing power, work, covered distance and time to exhaustion as well as reducing completion time"

The sentence has been changed as suggested.

Good paper that adds value to literature on the area.

Thank you very much for all of the suggestions.

Reviewer 2 Report

General comments

The authors have reviewed the current evidence for the effect of pre cooling techniques on athletic performance in highly trained athletes in hot conditions. They conclude that pre cooling techniques can be used by highly trained athletes in hot conditions to improve performance, and provide recommendations for the type of pre cooling techniques that should be used and the situations where athletes may benefit the most.

However, as it stand several major alterations are required to meet successful publication standards. The overall rationale for this review is weak and this needs to be more convincing. This is particularly an issue since other relatively recent and comprehensive reviews on a similar topic have been citied. It is not clear how different pre cooling techniques impact athletic performance in the results section, therefore it is not clear to the reviewer how the authors have arrived at their conclusion. The discussion of the results is adequate, although hard to follow, potentially due to there being little detail in the results section of the manuscript. Finally, the authors should proof read the manuscript carefully to identify missing words, grammatical errors and spelling mistakes as there are several throughout.

Introduction

1)      The introduction is concise and sets this review up nicely for an analyses in highly trained athletes. It is clear that pre cooling techniques have the potential to improve athletic performance, however there is no focus on the issues/dangers or heat illness associated with hyperthermia, only performance is discussed.

2)      Authors mentions of the benefits of cooling during performance, however it is unclear whether the focus of this review is the effect of cooling during or just before an event on performance outcomes.

3)      Given the immediate focus on performance in elite athletes some information acknowledging the differences compared to sendentary populations will strongly improve the overall rationale.

4)      The authors direct the reader to a number of other relatively recent and comprehensive reviews on this topic. These reviews may not have focused on highly-trained athletes alone, but the authors do not provide a rationale as to why these athletes may react differently to pre cooling techniques compared to athletes in these review articles. Without this it is not clear what this manuscript is adding to the knowledge on this topic.

Methods

1)      Clear outline of methods used and study inclusion criteria appears rigorous and valid

2)      Use of headings would make this section easier to follow

Results

1)      Headings required here this will help improve focus.

2)      The authors fail to clearly outline the results of the review in detail. Readers are directed to a large table which outlines the results of each study, however this does not make it easy for the reader to understand the effect of different pre cooling techniques on performance. The authors provide a brief statement on the effect of pre cooling in general on different aspects of performance, however much more can be gained from the results of this review in terms of understanding how different pre cooling techniques i.e. external, internal or mixed affect performance differently and which is more effective. The reviewer suggests including sub sections that outline to the reader the effect of these different pre cooling techniques on performance in greater detail.

3)      The table could be made easier to understand quickly, i.e. under type of pre cooling technique write EXT, INT or MIX depending on the type of technique used in the study.

4)      A more visual representation of the data e.g. a forest plot would summarize the results of the review much more clearly.

5)      Effect size calculations and 95% confidence intervals would provide better insight into the effect of the different pre cooling techniques on performance and how valuable these techniques are.

6)      The table that outlines the quality of each study is not relevant, a sentence in the discussion to outline the overall quality of studies that have addressed this question and potential areas for improvement in further research is sufficient.

Discussion

1)      There is adequate discussion of the effectiveness of individual pre cooling techniques in terms of performance and how pre cooling may affect performance beyond attenuation of heat stress (i.e. the detrimental effects of pre cooling on recruitment of fast twitch fibers during sprint exercise). The effect of event duration on the effectiveness of pre cooling is also discussed, which is an important consideration for athletes and coaches.

2)      The major findings are addressed in this section, however it is hard to follow with such little context from the results section.

3)      The authors appear to list studies rather than using the findings to explain their own results

4)      Line 152: Maroni at al. did not show any difference in what?

Future directions

1)      Authors should provide directions for researchers based on the discussion i.e. effect of pre cooling on short and intense efforts still unknown.

Conclusion

1)      The conclusions appear valid from information in the discussion, however how the authors came to this conclusion could be more clear if more information was included in the results. The authors make clear recommendations on the most effective pre cooling techniques so that the information can be easily applied by athletes and coaches.

Author Response

Comments and Suggestions for Authors

General comments

The authors have reviewed the current evidence for the effect of pre cooling techniques on athletic performance in highly trained athletes in hot conditions. They conclude that pre cooling techniques can be used by highly trained athletes in hot conditions to improve performance, and provide recommendations for the type of pre cooling techniques that should be used and the situations where athletes may benefit the most.

However, as it stands several major alterations are required to meet successful publication standards. The overall rationale for this review is weak and this needs to be more convincing. This is particularly an issue since other relatively recent and comprehensive reviews on a similar topic have been citied. It is not clear how different pre cooling techniques impact athletic performance in the results section, therefore it is not clear to the reviewer how the authors have arrived at their conclusion. The discussion of the results is adequate, although hard to follow, potentially due to there being little detail in the results section of the manuscript. Finally, the authors should proof read the manuscript carefully to identify missing words, grammatical errors and spelling mistakes as there are several throughout.

Thank you for your criticism. We have re-examined the whole manuscript and we have corrected some grammatical errors and spelling mistakes. We have made a quick search in the databases to confirm that there were not new studies to include up to April 2020.

Introduction

  • The introduction is concise and sets this review up nicely for an analyses in highly trained athletes. It is clear that pre cooling techniques have the potential to improve athletic performance, however there is no focus on the issues/dangers or heat illness associated with hyperthermia, only performance is discussed.

That’s right! We have added a sentence referring to the health disturbances derived from hyperthermia. Furthermore, exertional heat stroke may occur when core temperature reaches 40 °C and the subject begins to suffer changes in mental status [7].” (lines 45-46).

However, we have only discussed performance because, as we mention in the introduction, highly-trained athletes respond to heat as if they were already acclimatized. Based on this, we have focused on performance parameters.

2)      Authors mentions of the benefits of cooling during performance, however it is unclear whether the focus of this review is the effect of cooling during or just before an event on performance outcomes.

Sorry for the misunderstanding. We describe in the introduction that athletes can benefit from cooling, either pre- or during competition. We have named it in order to give a general approach of the issue. Then, we specified that we were going to focus only on pre-cooling techniques.

 “Therefore, the purpose of this systematic review was to summarize the current scientific evidence in relation to the effectiveness of pre-cooling strategies in highly-trained athletes exercising/competing in high temperature environments.” (lines 60-62).

3)      Given the immediate focus on performance in elite athletes some information acknowledging the differences compared to sedentary populations will strongly improve the overall rationale.

Thanks for your appreciation. We have tried to describe the differences through this sentence, in which we compared highly-trained athletes with moderately-trained or untrained individuals: “Highly-trained endurance athletes respond physiologically as if they were already heat acclimatized [7], and present less adaptive potential in comparison with moderately-trained athletes or untrained subjects [8]. In fact, a study performed in elite-cyclists competing in heat showed that they are able to reach core temperatures of 40 °C and above without heat illness [9].”(lines 46-50).

4)      The authors direct the reader to a number of other relatively recent and comprehensive reviews on this topic. These reviews may not have focused on highly-trained athletes alone, but the authors do not provide a rationale as to why these athletes may react differently to pre cooling techniques compared to athletes in these review articles. Without this it is not clear what this manuscript is adding to the knowledge on this topic.

We have re-examined the reviews we have cited and we proceed to establish the differences between them and our review.

  1. Wegmann et al. (2012):
  2. Includes studies performed in moderate temperatures (less than 30ºC).
  3. Does not specify if subjects were pre-acclimatized or not.
  4. We have 11 references in common.
  5. Bongers et al. (2017):
  6. Is not a systematic review.
  7. Includes studies performed in moderate temperatures (less than 30ºC)
  8. Includes both pre- (cooling prior exercise) and per-cooling (cooling during exercise) protocols.
  9. Does not specify level of fitness (VO2max)
  10. We have 6 references in common.
  11. Ross et al. (2013):
  12. Includes studies performed in moderate temperatures (less than 30ºC)
  13. Includes both pre- (cooling prior exercise) and per-cooling (cooling during exercise) protocols.
  14. Does not specify if subjects were pre-acclimatized or not.
  15. We have 10 references in common.
  16. Tyler et al. (2015):
  17. Includes studies performed in moderate temperatures (less than 30ºC)
  18. Includes both pre- (cooling prior exercise) and per-cooling (cooling during exercise) protocols.
  19. Includes only external cooling techniques (it eliminates internal and mixed methods).
  20. Does not specify level of fitness (VO2max)
  21. We have 9 references in common.
  22. Hohenauer et al. (2018):
  23. Includes both athletes (also moderately-trained) and healthy active volunteers.
  24. Includes studies performed in moderate temperatures (less than 30ºC).
  25. Does not specify if subjects were acclimatized to heat or not.
  26. We have 13 references in common.

Bearing in mind that none of the above reviews has addressed the topic according to the criteria we intended, we considered to perform a systematic review focusing on: 1) pre-cooling methods (both external, internal and mixed), 2) hot temperatures (>30ºC), 3) highly-trained individuals (>55 mL/kg/min of VO2max) who were heat-acclimatized.

Methods

1)      Clear outline of methods used and study inclusion criteria appears rigorous and valid

2)      Use of headings would make this section easier to follow

We have divided the methods section using headings (Search strategy; Eligibility criteria; Study selection; Quality assessment).

Results

1)      Headings required here this will help improve focus.

We have divided the results section through headings (Studies included; Participant characteristics; Intervention characteristics; Outcome measures [with subheadings]; Quality assessment).

  • The authors fail to clearly outline the results of the review in detail. Readers are directed to a large table which outlines the results of each study, however this does not make it easy for the reader to understand the effect of different pre cooling techniques on performance. The authors provide a brief statement on the effect of pre cooling in general on different aspects of performance, however much more can be gained from the results of this review in terms of understanding how different pre cooling techniques i.e. external, internal or mixed affect performance differently and which is more effective. The reviewer suggests including sub sections that outline to the reader the effect of these different pre cooling techniques on performance in greater detail.

We are aware that Table 1 has a lot of information because the review incorporates a large number of articles. We destined the results section to summarize and schematize the most striking characteristics of the studies and the most relevant results. Then, we have extended in the discussion, in with we have explained point-by-point each of the techniques, and we have concluded that mixed approaches (those which combine internal and external cooling methods) are the most effectiveness in terms of performance, following by cold water immersion.

3)      The table could be made easier to understand quickly, i.e. under type of pre cooling technique write EXT, INT or MIX depending on the type of technique used in the study.

Thank you for your recommendation. We agree with that and we have been made the change.

4)      A more visual representation of the data e.g. a forest plot would summarize the results of the review much more clearly.

5)      Effect size calculations and 95% confidence intervals would provide better insight into the effect of the different pre cooling techniques on performance and how valuable these techniques are.

We agree with your opinion. However, the primal objective of this review was not made a quantitative analysis of literature, but a qualitative approach. We intended to perform a qualitative synthesis on this topic, leading the reader to a quick map about the use of pre-cooling methods and its effects on sports performance. This approach is closer to that of a narrative review instead of a meta-analytic review, although it has the advantage of having been carried out in a systematic manner.

Furthermore, performing a statistical analysis would greatly modify the current structure of the manuscript, based on the potential results of that analysis. This fact would lead us to redo the manuscript almost completely (results and discussion sections), and it would be an inconvenient. In addition, it should be borne in mind that, once this analysis has been carried out, the results could contain a great level of heterogeneity that would make impossible establish solid conclusions.

Nevertheless, we are aware that the lack of a statistical analysis could be a limitation of our study.

6)      The table that outlines the quality of each study is not relevant, a sentence in the discussion to outline the overall quality of studies that have addressed this question and potential areas for improvement in further research is sufficient.

As far as we know, all the systematic reviews include a table with bias or methodological quality assessment. We have looked for two examples in this journal and they include a similar table. (Auyeung, A.B.; Almejally, A.; Alsaggar, F.; Doyle, F. Incidence of Post-Vasectomy Pain: Systematic Review and Meta-Analysis. Int. J. Environ. Res. Public Health 202017, 1788; Timme M, Bohner L, Huss S, Kleinheinz J, Hanisch M. Response of Different Treatment Protocols to Treat Chronic Non-Bacterial Osteomyelitis (CNO) of the Mandible in Adult Patients: A Systematic Review. Int J Environ Res Public Health. 2020;17(5):1737).

Discussion

1)      There is adequate discussion of the effectiveness of individual pre cooling techniques in terms of performance and how pre cooling may affect performance beyond attenuation of heat stress (i.e. the detrimental effects of pre cooling on recruitment of fast twitch fibers during sprint exercise). The effect of event duration on the effectiveness of pre cooling is also discussed, which is an important consideration for athletes and coaches.

2)      The major findings are addressed in this section, however it is hard to follow with such little context from the results section.

Based on your suggestion, we have enlarged the results section, and we have divided the “Outcome measures” heading in several subheadings according to cooling methods. This could facilitate the lector to organize his reading.

3)      The authors appear to list studies rather than using the findings to explain their own results

Once we have evaluated the results from the studies included in the review, we have looked for other studies with similar or different results in order to establish confrontations or comparisons between them. Bearing in mind that it is not a clinical trial, which results would be “our own” and could be explained with greater prominence, we have discussed results using a great variety of studies to enrich the review.

4)      Line 152: Maroni at al. did not show any difference in what?

That’s right. We refer to performance. We have added it (“Maroni et al [40] did not show any differences in performance when…”) (lines 204-207).

Future directions

1)      Authors should provide directions for researchers based on the discussion i.e. effect of pre cooling on short and intense efforts still unknown.   

We agree with your opinion and we have added a sentence explaining this issue. “We have concluded that pre-cooling is effectiveness in improve performance parameters in highly-trained athletes. However, no solid evidence exists about the implication of cooling strategies on short and intense efforts, as we have mentioned above. Furthermore, it is also important to find the most suitable pre-cooling technique for each sports practice or specific context, since it may occur that the most effective technique in absolute terms is not the most adequate in certain circumstances.” (lines 289-294).

Conclusion

  • The conclusions appear valid from information in the discussion, however how the authors came to this conclusion could be more clear if more information was included in the results. The authors make clear recommendations on the most effective pre cooling techniques so that the information can be easily applied by athletes and coaches.

Thank you very much for your comments.

Round 2

Reviewer 2 Report

Thank you to the authors for their edits and comments. Happy to see this published.